# Structural anatomy of Protein Kinase C C1 domain interactions with diacylglycerol and other agonists

Sachin S. Katti[1], Inna V. Krieger [1], Jihyae Ann [2], Jeewoo Lee [2], James C. Sacchettini [1] & Tatyana I. Igumenova [1✉]

Diacylglycerol (DAG) is a versatile lipid whose 1,2-*sn*-stereoisomer serves both as second messenger in signal transduction pathways that control vital cellular processes, and as metabolic precursor for downstream signaling lipids such as phosphatidic acid. Effector proteins translocate to available DAG pools in the membranes by using conserved homology 1 (C1) domains as DAG-sensing modules. Yet, how C1 domains recognize and capture DAG in the complex environment of a biological membrane has remained unresolved for the 40 years since the discovery of Protein Kinase C (PKC) as the first member of the DAG effector cohort. Herein, we report the high-resolution crystal structures of a C1 domain (C1B from PKCδ) complexed to DAG and to each of four potent PKC agonists that produce different biological readouts and that command intense therapeutic interest. This structural information details the mechanisms of stereospecific recognition of DAG by the C1 domains, the functional properties of the lipid-binding site, and the identities of the key residues required for the recognition and capture of DAG and exogenous agonists. Moreover, the structures of the five C1 domain complexes provide the high-resolution guides for the design of agents that modulate the activities of DAG effector proteins.

[1] Department of Biochemistry and Biophysics, Texas A&M University, College Station, TX 77840, USA. [2] College of Pharmacy, Seoul National University, Seoul 08826, Republic of Korea. ✉email: Tatyana.Igumenova@ag.tamu.edu

The impressive diversity of DAG signaling output is mediated via its interactions with seven families of effector proteins that execute broad sets of regulatory functions[1]. These include protein phosphorylation (PKCs and PKDs[2,3]); DAG phosphorylation (DGKs[4]); RacGTPase regulation (Chimaerins[5]); Ras guanine nucleotide exchange factor activation (RasGRPs[6]); Cdc42-mediated cytoskeletal reorganization (MRCK[7]); and assembly of scaffolds that potentiate synaptic vesicle fusion and neurotransmitter release (Munc13s[8]). PKCs define a central DAG-sensing node in intracellular phosphoinositide signaling pathways that regulate cell growth, differentiation, apoptosis, and motility[9]. Shortly after their discovery, PKCs were identified as cellular receptors for tumor-promoting phorbol esters[10] that bind C1 domains in lieu of DAG. These observations, combined with the central roles executed by PKCs in intracellular signaling established their DAG-sensing function as an attractive target for therapeutic intervention, with considerable promise in the treatment of Alzheimer's disease[11], HIV/AIDS[12,13], and cancer[14,15]. However, the structural basis of DAG recognition by the C1 domains has remained elusive, and the strategies for therapeutic agent design deployed to date all relied on modeling studies (reviewed in[16]) based on the single available crystal structure of the C1 domain complexed to a ligand that does not activate PKC[17]. Herein, we have overcome these well-documented challenges[18–20] that have hindered the crystallization of extremely hydrophobic C1-ligand complexes for almost three decades. This advance enabled us to determine high-resolution structures of C1 bound to the endogenous agonist DAG and to each of four exogenous agonists of therapeutic interest. Collectively, our findings: (i) provide a structural rationale for the consensus amino acid sequence of DAG-sensitive C1 domains; (ii) provide insight into the origins of DAG sensitivity; and (iii) reveal how the unique hydrophilic/hydrophobic properties of the ligand-binding groove enable C1 domains to accommodate chemically diverse ligands.

## Results and discussion

**Structure of the C1Bδ-DAG complex.** C1 domains form complexes with their ligands in the membrane environment, and dodecylphosphocholine (DPC) is an effective membrane mimic that faithfully reproduces the functional properties of C1 domains with respect to ligand-binding interactions[21,22]. For structural studies, we formed the complex between the C1B domain of PKCδ (C1Bδ) and a synthetic DAG analog (di-octanoyl-sn-1,2-glycerol) in the presence of DPC micelles. The structure of the C1Bδ-DAG complex was refined to 1.75 Å with $R_{work} = 0.214$ and $R_{free} = 0.246$ (Supplementary Table 1). The H3 space group unit cell ($89 \times 89 \times 219$ Å, Fig. 1a) contains a total of 72 protein chains contributed by nine asymmetric units (AUs) with eight C1Bδ molecules per AU (Fig. 1b).

The organization of the protein crystal lattice is unprecedented as it contains two distinct lipid-detergent micelles located at symmetry elements in the crystal. Micelle 1 is composed of 12 DAG and 12 DPC ordered molecules, and micelle 2 composed of 18 DAG and 6 DPC ordered molecules (Fig. 1c), in addition to fully or partially disordered lipids. We speculate that the micelles help nucleate the crystallization, as all of the protein subunits are arranged with their lipid sensing loops directly binding to DAG within micelles (Supplementary Fig. 1a, b). Each C1Bδ protein chain has a DAG molecule captured within a groove formed by its membrane-binding loop regions (Fig. 1b and Supplementary Fig. 1c, d). The well-defined glycerol ester moieties of this tightly bound 'intra-loop' DAG refined with B-factors (18–26 Å²) comparable to those of the surrounding protein residues (17–23 Å²). In addition, each AU contains two less-ordered

peripheral DAG and six DPC molecules associated with the amphiphilic protein surface (Supplementary Fig. 1e, f).

There is little variability among DAG-complexed C1Bδ chains within the asymmetric unit, as evidenced by low pairwise backbone r.m.s.d values of 0.3–0.6 Å (Fig. 2a). The most variable region is located between helix α1 and the C-terminal Cysteine residue that coordinates a structural $Zn^{2+}$ ion. According to solution NMR, this region undergoes conformational exchange on the µs-ms timescale[21,23]. We crystallized apo C1Bδ under conditions similar to those used for the C1Bδ-DAG complex (but without detergent) for a direct comparison. We found the structure to be identical to the previously reported apo structure (1PTQ[17]) with a backbone r.m.s.d. of 0.4 Å. The apo structure superimposes well onto the structures of DAG-complexed C1Bδ, with the notable exception of the Trp252 sidechain. In the DAG complex, this sidechain is oriented towards the DAG tethered to the membrane-binding region, whereas in the apo C1Bδ it is oriented away from that region (Fig. 2a). This was a satisfying result as Trp252 is associated with the "DAG-toggling" behavior of the C1 domains, wherein a conservative Trp→Tyr substitution in the novel (or $Ca^{2+}$-insensitive) and Tyr→Trp substitution in conventional (or $Ca^{2+}$-activated) PKC isoforms significantly modulates apparent affinity for DAG[21,22,24,25].

**Stereospecificity of DAG binding by C1Bδ.** Another significant feature of the C1Bδ-DAG structure is that it reveals the mechanism for the stereospecific binding of sn-1,2-diacylglycerol by C1 domains. DAG binds in the groove formed by the protein loops, β12 and β34 (Fig. 2a) and can adopt two distinct binding modes: "sn-1" and "sn-2"[26] (Fig. 2b, c). The "sn-1" mode is predominant in the crystal as it is observed in six of the eight C1Bδ chains (Supplementary Fig. 2a). We note that the DAG/DPC micelle 1 exclusively supports the "sn-1" binding mode, while in micelle 2, both "sn-1" and "sn-2" interactions are found (Supplementary Fig. 1b).

In both modes, the DAG glycerol and ester moieties are anchored to the C1Bδ binding groove by four hydrogen bonds. Three are contributed by the C3-OH hydroxyl group that serves as the donor for the carbonyl oxygens of Thr242 and Leu251, and as the acceptor for the amide hydrogen of Thr242. The fourth bond involves the amide hydrogen of Gly253 and it is this bond that defines the binding mode. In the "sn-1" position, the acceptor is the carbonyl oxygen O5 of the sn-1 ester group, whereas in the "sn-2" position the acceptor is the carbonyl oxygen O3 of the sn-2 ester group (Fig. 2b, c).

A particularly important feature of the "sn-1" binding mode is the involvement of the alkoxy oxygen O2 of the sn-2 DAG chain in the hydrogen bond with the amide protons of the Gln257 sidechain (Fig. 2b, Supplementary Fig. 2b). Gln257 is part of the strictly conserved "QG" motif in all DAG-sensitive C1 domains[27] that is essential for agonist binding[19,21,28,29], and whose µs-ms dynamics in apo C1 domains correlates with DAG-binding affinity[23]. The Gln257 sidechain also "stitches" the β12 and β34 loops together, forming hydrogen bonds with Tyr238 of loop β12 and Gly253 of loop β34 (Supplementary Fig. 2b). The simultaneous involvement of Gln257 in both DAG and intra-protein stabilizing interactions explain its essential role in the formation of the C1Bδ-DAG complex.

**C1Bδ binds DAG in the "sn-1" mode in the solution.** To determine the predominant DAG-binding mode in solution, we conducted solution NMR experiments on C1Bδ-DAG assembled in isotropically tumbling bicelles. Complex formation is evident from the chemical shift perturbations of the C1Bδ backbone NH and the Trp252 NHε groups (Fig. 2d and Supplementary Fig. 3a), and of the methyl groups of hydrophobic residues residing in the loop regions

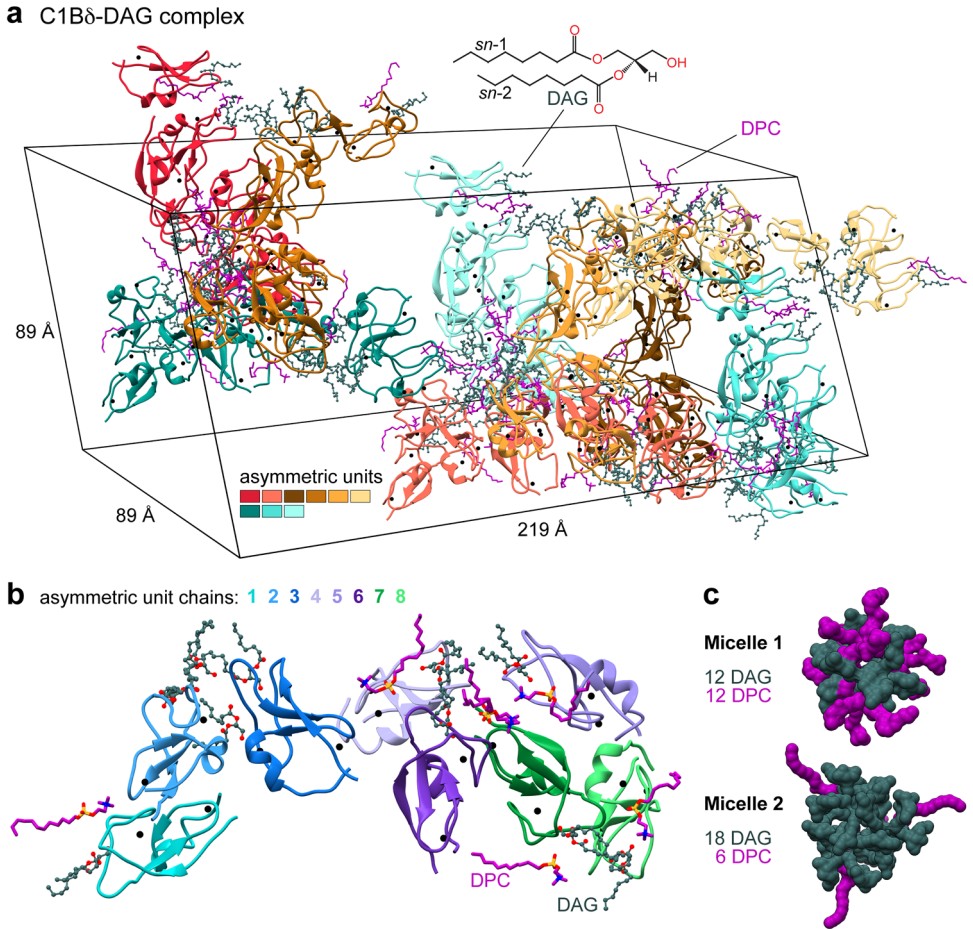

**Fig. 1 Arrangement of protein chains, lipid, and detergent molecules in the unit cell and the asymmetric unit of the C1Bδ-DAG complex crystal (PDB ID: 7L92). a** The unit cell contains 72 DAG-complexed C1Bδ chains and 18/54 DAG/DPC molecules that peripherally associate with the protein surface. Structural $Zn^{2+}$ ions of C1Bδ are shown as black spheres. **b** The asymmetric unit comprises 8 C1Bδ protein chains with 8 DAG molecules captured within a well-defined groove, and 2/6 peripheral DAG/DPC molecules. **c** Space-filling representation of the two distinct DAG/DPC micelles.

(Fig. 2e and Supplementary Fig. 3b). Also evident is rigidification of the loops upon DAG binding as manifested by the appearance of backbone NH cross-peaks broadened in the apo-state due to their intermediate-timescale loop dynamics (Supplementary Fig. 3a). 3D $^{15}N$-edited [$^1H$,$^1H$] NOESY experiments were performed where C1Bδ was extensively deuterated at the non-exchangeable sites to suppress intra-protein NOEs. The two DAG-binding modes observed in the crystal structure are associated with drastically different protein-to-ligand NOE patterns. The signatures of the "sn-1" mode are predicted to be short-to-medium range NOEs between $^1H_N$(Gly253) and $^1H_{CH2}$(C1), and a long-range NOE between $^1H_N$(Gly253) and $^1H_{CH2}$(C3). This is precisely the pattern observed experimentally in the Gly253 strip (Fig. 2f). The medium-range NOE between $^1H_N$(Gly253) and $^1H_{CH}$(C2) that would signify the "sn-2" mode (shown in red in the "sn-2" complex, Fig. 2f) is not detected. Further confirmation of the "sn-1" DAG-binding mode is provided by the Thr242 strip whose $^1H_N$ shows a characteristic medium-range NOE to $^1H_{CH}$(C2) and a short-range NOE to $^1H_{CH2}$(C3). Thus, both the C1Bδ-DAG structure and solution NMR experiments report "sn-1" as the dominant mode of DAG binding to C1Bδ. Moreover, also consistent with the C1Bδ-DAG structure, several C1Bδ loop residues (including Gly253) show NOEs to the methylenes of acyl chains of either DAG or bicelle lipids (Supplementary Fig. 3c, d).

**Roles of C1Bδ loops in lipid binding**. Inspection of C1Bδ loop regions in the C1Bδ-DAG structure reveals how exquisitely they

are tuned to the chemical properties of diacylglycerol and surrounding lipids. The eleven C1Bδ residues involved in DAG interactions (Supplementary Fig. 4a, b) can be grouped into three tiers that progressively increase the hydrophobicity of the ligand environment – as illustrated using the deconstructed binding groove of the representative "sn-1" complex (Fig. 3a). The atoms from four "tier 1" residues, namely the backbone O and NH groups of Tyr238, Thr242, and Leu251, along with the sidechain of Gln257, define the polar surface of the groove floor that accommodates the C3-OH hydroxyl group of DAG. The hydrophobic sidechains of tier 1 residues point towards the core of the protein and away from the groove. The five "tier 2" residues accommodate the glycerol backbone and ester groups of DAG by creating an amphiphilic environment. Met239 and Ser240 backbone O and N atoms provide a polar environment for the O3 oxygen of DAG, while their sidechains face "outward" to potentially engage in lipid interactions. Pro241, a strictly conserved residue in DAG-sensitive C1 domains, makes non-polar contacts with the C2 carbon of DAG, and its Cδ/Cα are positioned sufficiently close to the DAG oxygens to form C–H…O interactions (Supplementary Fig. 4c). On the opposite side of the groove, the polar N–H group of Gly253 hydrogen bonds to the O5 oxygen, while the hydrophobic Leu250 sidechain engages in non-polar contacts with the C1 carbon of DAG.

While tier 1 and 2 residues are contributed by both loops, tier 3 residues all reside on loop β34 (Fig. 3a). Trp252 and Leu254 sidechains make non-polar contacts with the methylenes of the

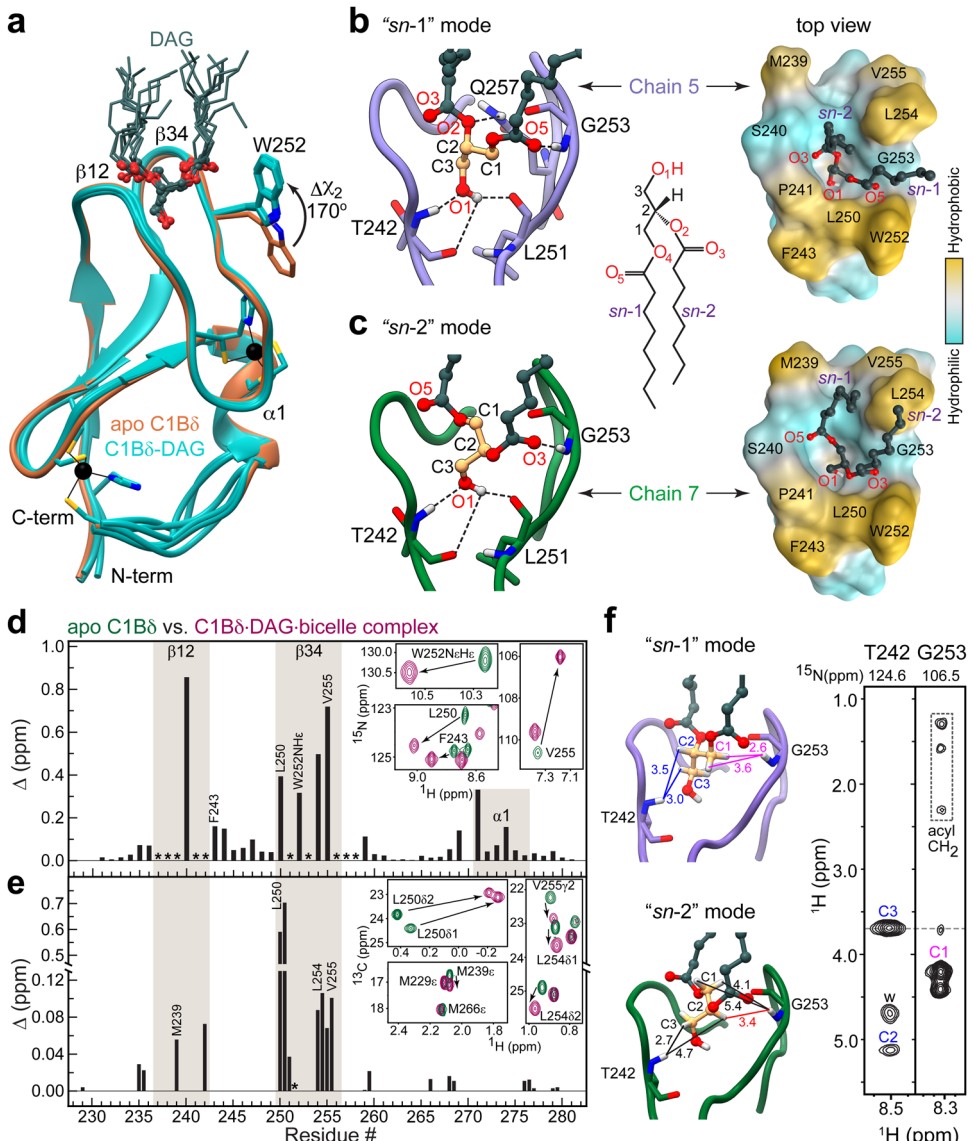

**Fig. 2 Stereospecificity of DAG binding by C1Bδ. a** Backbone superposition of 8 DAG-complexed C1Bδ chains of the AU (cyan, PDB ID: 7L92) onto the structure of apo C1Bδ (sienna, PDB ID: 7KND). The sidechain of Trp252 reorients towards the tips of membrane-binding β12 and β34 loops upon DAG binding. DAG adopts one of the two distinct binding modes: "*sn*-1" (**b**) or "*sn*-2" (**c**). The formation of the C1Bδ-DAG complex in bicelles is reported by the chemical shift perturbations (CSPs) of the amide $^{15}NH$ (**d**) and methyl $^{13}CH_3$ (**e**) groups of C1Bδ. Asterisks denote residues whose resonances are broadened by chemical exchange in the apo-state. The insets show the response of individual residues to DAG binding through the expansions of the $^{15}N$–$^1H$ and $^{13}C$–$^1H$ HSQC spectral overlays of apo and DAG-complexed C1Bδ. **f** $^1H$–$^1H_N$ Thr242 and Gly253 strips from the 3D $^{15}N$-edited NOESY-TROSY spectrum of the C1Bδ-DAG-bicelle complex. The protein-to-DAG NOE pattern is consistent with the distances observed in the "*sn*-1" mode (Chain 5, light purple) but not the "*sn*-2" mode (Chain 7, green). All distances are in Å and color-coded in the "*sn*-1" complex to match the labels in the spectrum; "w" denotes water protons. The medium-range NOE that would be characteristic of the "*sn*-2" complex is shown in red.

DAG *sn*-1 and *sn*-2 acyl chains (Supplementary Fig. 4a) that protrude through the depression formed by Gly253 (Figs. 2b, 3a). These interactions orient the DAG acyl chains in a position to complete the hydrophobic rim of the C1Bδ domain formed by loop β34 residues Leu250, Trp252, Leu254, Val255, and loop β12 residues Met239 and Pro241. This is illustrated in the top views of the "*sn*-1" and "*sn*-2" C1Bδ-DAG complexes (right panels of Fig. 2b, c). Positioning the acyl chains in close proximity to Trp252 and Leu254 sidechains creates a continuous hydrophobic surface tailored for C1Bδ interactions with surrounding lipids. In this context, the significance of Trp252 sidechain reorientation upon DAG binding (Fig. 2a) becomes clear, as this conformation ensures the continuity of the hydrophobic surface. A direct manifestation of the lipophilicity of the DAG-bound C1Bδ surface

is the peripheral association of DAG and detergents observed in all crystal structures of the complexes (Supplementary Figs. 5, and 6; Supplementary Note 2).

To directly identify the regions of the C1Bδ-DAG that insert into the bilayer and to independently validate the essential role of the β34 loop in membrane partitioning, paramagnetic relaxation enhancement (PRE) experiments were performed using a paramagnetic lipid (14-doxyl PC) incorporated into host bicelles. PREs arise due to the spatial proximity of the unpaired electron of the lipid probe to protein nuclear spins and manifest themselves as extensive line broadening in the NMR spectra. PRE data for the C1Bδ-DAG amide hydrogens report that, while both loops undergo bilayer insertion, loop β34 penetrates deeper into the bilayer than does loop β12 (Fig. 3b). The PRE data are entirely

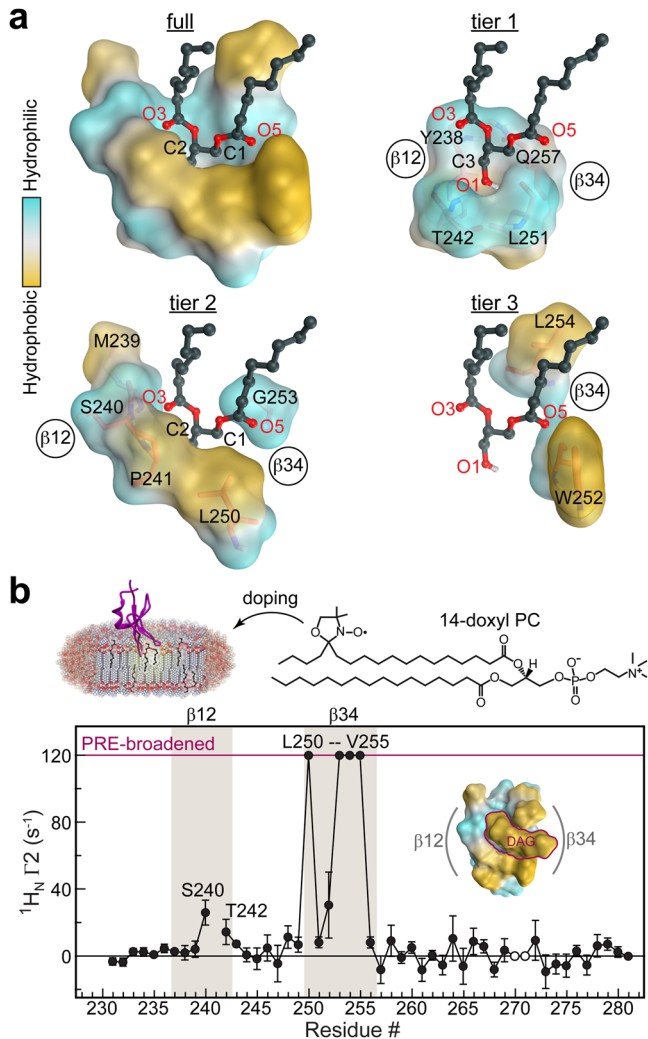

**Fig. 3 Roles of C1Bδ loops in lipid binding. a** Polar backbone atoms and hydrophobic sidechains of DAG-interacting C1Bδ residues create a binding site whose properties are tailored to capture the amphiphilic DAG molecule. This is illustrated through the deconstruction of the "*sn-1*" binding mode into three tiers that accommodate the glycerol backbone (tier 1), the *sn-1/2* ester groups (tier 2), and the acyl chain methylenes (tier 3). **b** Residue-specific lipid-to-protein PRE values of the amide protons, $^1H\ \Gamma_2$, indicate that loop β34 is inserted deeper into the membrane than β12. The PRE value for Trp252 is that for the indole NHε group. Cross-peaks broadened beyond detection in paramagnetic bicelles are assigned an arbitrary value of $120\ s^{-1}$. His270 and Lys271 cross-peaks are exchange-broadened and therefore unsuitable for quantitative analysis (open circles). The PRE values were derived from $^1H_N$ transverse relaxation rate constants collected on a single sample in the absence (diamagnetic) and presence (paramagnetic) of 14-doxyl PC. The error was estimated using the r.m.s.d. of the base plane noise. The inset shows the top view of the "*sn-1*" mode C1Bδ-DAG complex color-coded according to the hydrophobicity.

consistent with the hydrophobicity patterns observed in the crystal structure (Fig. 3b, inset), and project that C1Bδ assumes a tilted position relative to the membrane normal upon DAG binding.

Of note, the deconstructed binding groove of the "*sn-2*" complex shows a similar three-tiered arrangement of hydrophilic and hydrophobic features equally suited to accommodate DAG in the "*sn-2*" orientation (Supplementary Fig. 4b, d). However, because of the differences in the hydrogen-bonding patterns (left panels of Fig. 2b, c) DAG position is shallow compared to the "*sn-1*" mode,

where the DAG C1 carbon resides deeper in the pocket by ~1.5 Å. While our data support the "*sn-1*" as the dominant DAG interaction mode, the presence of the "*sn-2*" mode in the crystal structure (Supplementary Figs. 1b, 2a) suggests that it too is sampled transiently during the DAG capture step.

**C1Bδ complexes with exogenous PKC agonists**. The DAG-sensing function has been an active target for pharmaceutical modulation of PKC activity. To determine the structural basis of how C1 domains mediate the response of DAG effector proteins to potent exogenous agonists, we selected four such agonists that evoke distinct cellular responses. Phorbol 12,13-dibutyrate (PDBu) is one of the potent tumor-promoting phorbol esters widely used to generate carcinogenesis models through PKC dysregulation[30]. Prostratin, a non-tumorigenic phorbol ester, is a preclinical candidate for inducing latency reversal in HIV-1 infection[31,32]. Both PDBu and Prostratin share a tetracyclic tigliane skeleton of 5-7-6-3 membered rings. Ingenol-3-angelate (I3A) is a clinically approved agent for the topical treatment of actinic keratosis (Picato®) with a phorbol-related 5-7-7-3 fused ring structure[33]. AJH-836 is a high-affinity synthetic DAG lactone with considerable promise as an isoenzyme-specific PKC agonist (Supplementary Note 3)[34]. In the membrane-mimicking lipid bicelle environment, all four ligands bind to the loop region of C1Bδ—as evidenced by the chemical shift perturbations and rigidification of the corresponding residues (Supplementary Fig. 7). Crystals of each C1Bδ-ligand complex formed in the presence of 1,2-diheptanoyl-*sn*-glycero-3-phosphocholine (DHPC) and all yielded high-resolution structures (1.1–1.8 Å, Supplementary Tables 1, 2, Fig. 4) with well-defined electron densities of ligands (Supplementary Fig. 8a). DHPC molecules peripherally associate with the protein surface outside loop β12 (I3A complex), loop β34 (Prostratin and AJH-836 model 7LF3), or both loops (PDBu and ligand-free chain of the AJH-836 model 7LEO) (Fig. 4a–e). The DHPC-protein interactions involve hydrogen bonds, both direct and water-mediated, as well as hydrophobic contacts. The structure of the AJH-836, where one protein chain has a bound ligand and the other ligand-free chain interacts peripherally with DHPC, highlights the versatility of interactions that Trp252 at the "DAG-toggling" position can form with lipids (Fig. 4e).

The structures reveal that all four ligands (Fig. 5a) bind to the same C1Bδ site as DAG (Fig. 5b). However, none of them form a hydrogen bond with the Gln257 sidechain that is observed in the DAG complex structure (Fig. 2b). The fused ring structures of PDBu, Prostratin, and I3A intercalate between loops β12 and β34. The methyl groups attached to rings A, C, and D, and the apical regions of rings A and C (Prostratin and I3A only) protrude outwards from the groove (Supplementary Fig. 8b). These groups collectively form a hydrophobic ridge that traverses the groove diagonally from Met239 of loop β12 to Trp252 of loop β34 (Fig. 5b). The AJH-836 lactone ring also intercalates between the loops and is fully sequestered.

The arrangement and identity of the R groups provide the unique shape of the hydrophobic "cap" over the loop region (Fig. 5b, surface representation). In the PDBu complex, the butyryl R1 and R2 groups are arranged in a T-shape relative to the ridge. Prostratin, having only a single acetyl R2 group, forms a significantly smaller hydrophobic cap. Neither ligand covers the depression in the β34 loop formed by Gly253—thereby creating an opportunistic interaction site for DHPC molecules in the crystal (Figs. 4a, b, and 5c). In contrast, bound I3A and AJH-836 form a contiguous hydrophobic surface with loop β34, while leaving loop β12 exposed and available for potential interactions with lipids (Figs. 4c and 5b). The angelyl and pivaloyl R1 groups

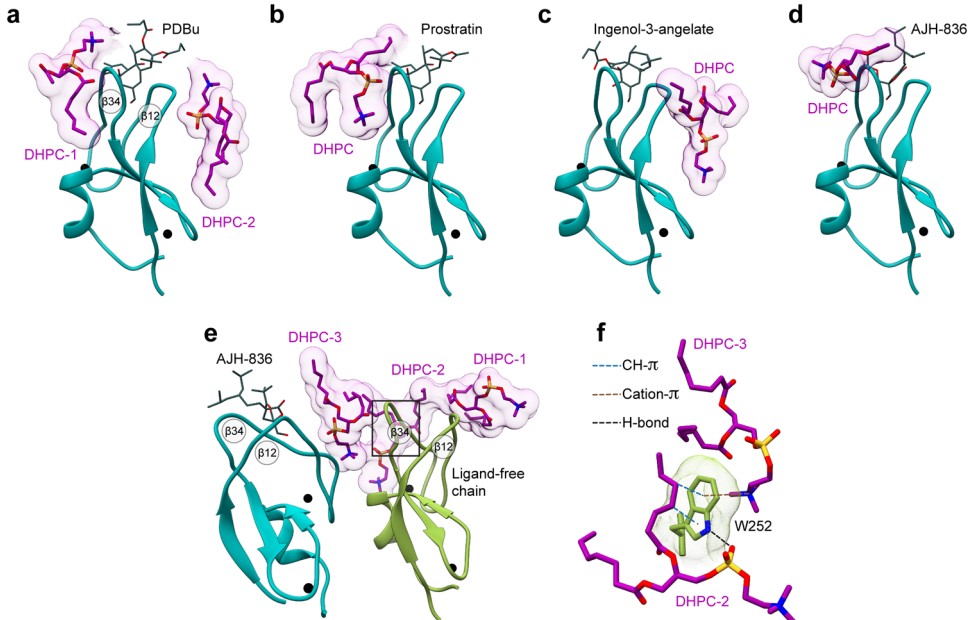

**Fig. 4 Peripheral DHPC molecules in the C1Bδ-ligand complexes.** DHPC molecules peripherally associate with the membrane-binding loop regions of C1Bδ complexed to **a** PDBu; **b** prostratin; **c** ingenol-3-angelate, **d** AJH-836 (one molecule per AU), and **e** AJH-836 (two molecules per AU). In **e**, one protein chain is ligand-free (color-coded green) and has three DHPC molecules, labeled 1 through 3 that cap the membrane-binding loop region. **f** The versatility of potential Trp252-lipid interactions, exemplified by the Trp252 sidechain from the ligand-free C1Bδ monomer (**e** green). In addition to non-polar contacts with the hydrophobic lipid moieties, the Trp sidechains can engage in H-bonding, cation–π, and CH–π interactions.

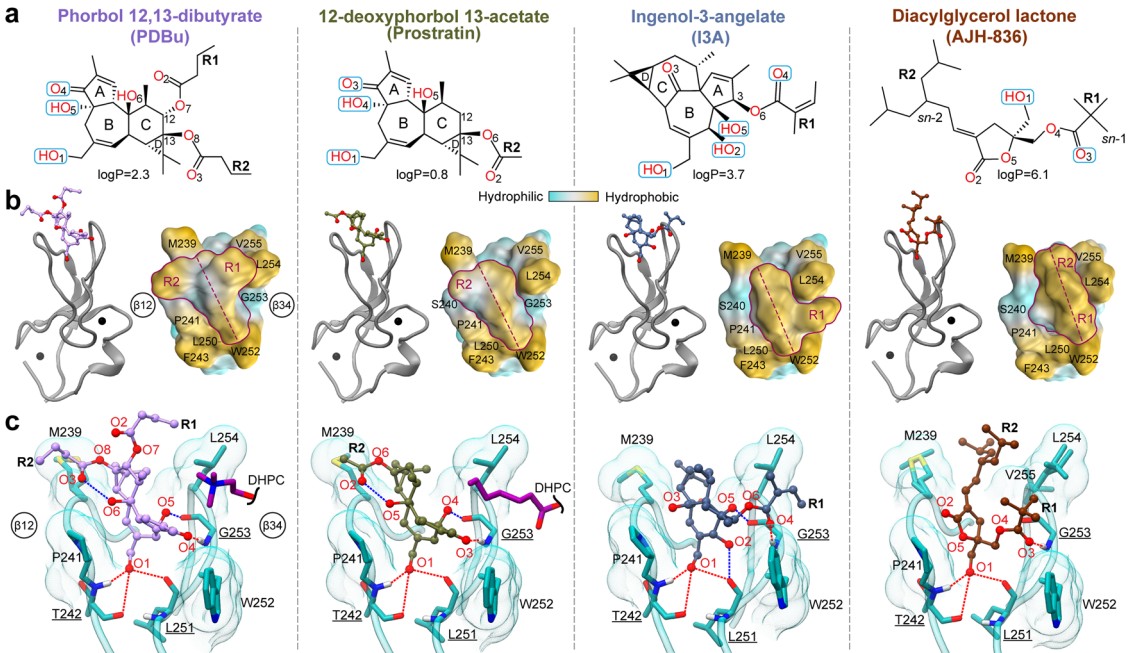

**Fig. 5 Structures of the C1Bδ-ligand complexes reveal the interaction modes of PKC agonists. a** Chemical structures and polar groups involved in hydrogen-bonding interactions with C1Bδ of PKC agonists. The numbering of oxygen atoms follows the ALATIS system. **b** 3D structures of the complexes (PDB IDs from left to right: 7KNJ, 7LCB, 7KO6, and 7LF3) showing the ligand placement in the binding groove. The shape of the ligands' hydrophobic cap, viewed from the top of the loop region, is outlined in maroon. The hydrophobic ridge that traverses the groove is marked with the maroon dashed line. **c** Ligand interactions with Thr242, Leu251, and Gly253 (underlined) that recapitulate the DAG hydrogen-bonding pattern are shown with red dashed lines. Blue dashed lines show ligand-specific hydrogen bonds, including the intra-ligand ones in PDBu and Prostratin. The depression created in loop β34 by Gly253 in the PDBu and Prostratin complexes accommodates DHPC molecules in the crystal.

of I3A and AJH-836, respectively, occupy the depression created by Gly253 and engage in hydrophobic contact with the sidechains of the bracketing residues Leu254 and Trp252 (Supplementary Fig. 8c). This arrangement presents as an overall L-shape of the I3A hydrophobic cap. The hydrophobic surface of AJH-836 comprises highly exposed methyl groups of the pivaloyl R1 and the branched alkylidene R2 groups (Supplementary Fig. 8b) that create a ridge at an ~20° angle with the long axis of the groove. Thus, each ligand modulates the shape and the hydrophobicity of the C1Bδ membrane-binding region in its own unique way by engaging its R groups in hydrophobic interactions with the same set of protein residues (Met239, Leu250, Trp252, Leu254, and V255).

The hydrophilic groups of the ligands orient towards the polar regions of the groove and recapitulate the DAG hydrogen-bonding pattern: the carbonyl oxygen (O3 in Prostratin and AJH-836; O4 in PDBu and I3A) and the hydroxyl group O1-H form hydrogen bonds with Gly253, Thr242, and Leu251 (Fig. 5c, red dashed lines). In addition, Prostratin and PDBu engage a second hydroxyl group (O4-H and O5-H, respectively) in hydrogen bonding with the carbonyl oxygen of Gly253. In I3A, there are two additional hydroxyls (O2-H and O5-H) that hydrogen bond to the carbonyl oxygens of Leu251 and Gly253, respectively (Fig. 5c, blue dashed lines). AJH-836 is the only ligand of the four with no additional ligand-protein hydrogen bonds compared to DAG.

The PDBu and Prostratin structures explain the findings of the previous structure-activity studies of phorbol ester derivatives[20,35,36] that identified the essential role of the hydrophobic substituent, R1 at position C-12, in the PKC membrane insertion and activation. The increased potencies[35] of 12,13-di- and 12-mono-esters with hydrophobic substituents (e.g., PDBu and phorbol 12-myristate 13-acetate (PMA)) relative to 12-deoxyphorbol esters (e.g., Prostratin) can be ascribed to two factors. First, the C-12 R1 group complements the hydrophobic rim of loop β34 which is the primary driver of C1 membrane binding. Second, both R1 and R2 groups are involved in direct interactions with lipids, and thereby contribute to the stabilization of the membrane complex. Molecular dynamics simulations of the C1-ligand complexes suggest that both the chemical identity of the R1 group[37,20] and of the ligand itself (PMA vs. DAG)[37] influence the depth of C1 membrane insertion.

The I3A complex provides a structural rationale for the relative potencies of ingenol derivatives reported in the HIV-1 latency reversal studies[13]. The most potent ingenols exhibit conformationally restricted R1 substituents that can be accommodated in the depression formed by Gly253 and bracketed by Trp252 and Leu254. In the lactone complex, a combination of the *E* isomer and its "sn-1" binding mode (Supplementary Note 3) affords a favorable arrangement of bulky R1 and R2 groups within the hydrophobic rim of C1Bδ. This configuration likely constitutes the structural basis for why AJH-836 displays its marked selectivity for novel versus conventional PKC isoforms[34].

**Comparative analyses of the C1Bδ-agonist structures**. Comparative analyses of our DAG- and ligand-bound structures (Fig. 6) demonstrate why DAG-sensing C1 domains are capable of binding chemically diverse ligands with high affinity—a property that is driving the design of pharmacological agents. The amphiphilic binding groove, with progressively increasing hydrophobicity towards the rim of the membrane-binding region, is "tuned" to accommodate ligands with matching properties. The placement of three oxygen-containing groups, highlighted in Fig. 6b–e, ensures that the ligand is anchored to the polar groove regions. The ring structure invites intercalation between the C1

membrane-binding loops, while the hydrophobic substituents that protrude outward from the groove, akin to the DAG acyl chains, contribute to the membrane anchoring of the complex.

The comparative analysis also enables the assignment of specific functional roles to the residues of the C1 domain consensus sequence (Fig. 6f and Supplementary Note 1)[27]. Two groups of residues are of particular significance. The first group consists of the four strictly conserved non-$Zn^{2+}$ coordinating residues: Pro241, Gly253, Gln257 that directly interact with DAG (Figs. 2b, c, 3a, Supplementary Fig. 4a, b, d); and Gly258 which ensures conformational flexibility of the β34 loop (Supplementary Note 1). The second group comprises three hydrophobic residues: Leu250, Trp252, and Leu254 of loop β34 that show the deepest membrane insertion (Fig. 3b). Together with strictly conserved Pro241 and the consensus aromatic residue Phe243, these three residues form the outside hydrophobic "cage" that surrounds the various bound ligands (Fig. 6g, h). The spatial arrangement of the cage residues not only shields the hydrophilic ligand moiety from the hydrophobic membrane environment (Figs. 2b, c, and 4) but also enables the loop region of C1Bδ to effectively interface with peripheral lipids. The latter function is aptly exemplified by Trp252, whose highly lipophilic indole sidechain reorients towards the loop region upon the formation of C1-ligand complexes in the membrane-mimicking environment. Given that all C1 complexes reported in this work are with potent PKC agonists, we posit that the reorientation of the Trp252 in C1Bδ (Fig. 2a) is an essential aspect of the overall mechanism of membrane recruitment and agonist capture. Indeed, our previous work suggests that the Trp252 sidechain reorients towards the membrane-binding loops upon initial partitioning of C1Bδ into the hydrophobic environment prior to the agonist binding[22]. Once this "pre-DAG" C1 complex[38] is formed, the Trp sidechain plays an important role in the formation of the ligand-binding site through the completion of the hydrophobic "cage" (Fig. 6g, h). Therefore, the structural change associated with the Trp252 flip underlies two processes that take place in the hydrophobic environment—membrane partitioning and binding of the membrane-embedded ligand. Consistent with this notion, no rotameric flip was observed in the previously determined structure of the C1Bδ-phorbol-13-acetate (P13A) complex that was obtained under crystallization conditions lacking membrane mimics (Fig. 6i)[17]. P13A is an extremely weak agonist of PKC[39,40] that differs from Prostratin with respect to a single hydroxyl group at the C-12 position. This OH group is ~10 Å away from Trp252 and is unlikely to influence the sidechain conformation directly (Fig. 6i). Rather, it imparts a significant hydrophilic character onto the "cap" formed by the ligand over the loop regions of C1Bδ (Fig. 6j). Given the relative hydrophilicity of the ligand (logP of P13A is 0.2, as compared with Prostratin's 0.8), and of the complex itself, there is no thermodynamic incentive for the former to partition into the membranes in a process that involves the Trp252 sidechain reorientation.

In addition to its dual role in membrane recruitment and ligand capture, the Trp252 conformation, and interaction patterns are directly relevant to the question of DAG sensitivity of PKC isoforms—i.e., the parameter that defines the intrinsic thresholds of DAG-mediated activation (Supplementary Note 2). Our structural data (supported by previous NMR work[21–23]) suggest that higher hydrophobicity and lipophilicity of Trp confers thermodynamic advantages and hence higher DAG sensitivity to the C1B domains of novel PKC isoforms relative to conventional PKC isoforms that carry a Tyr at the equivalent position.

The atomistic details of our high-resolution structures of five C1Bδ-ligand complexes, particularly with regard to the arrangement of the ligand hydrophobic substituents within the binding groove and assignment of specific functional roles to the key C1

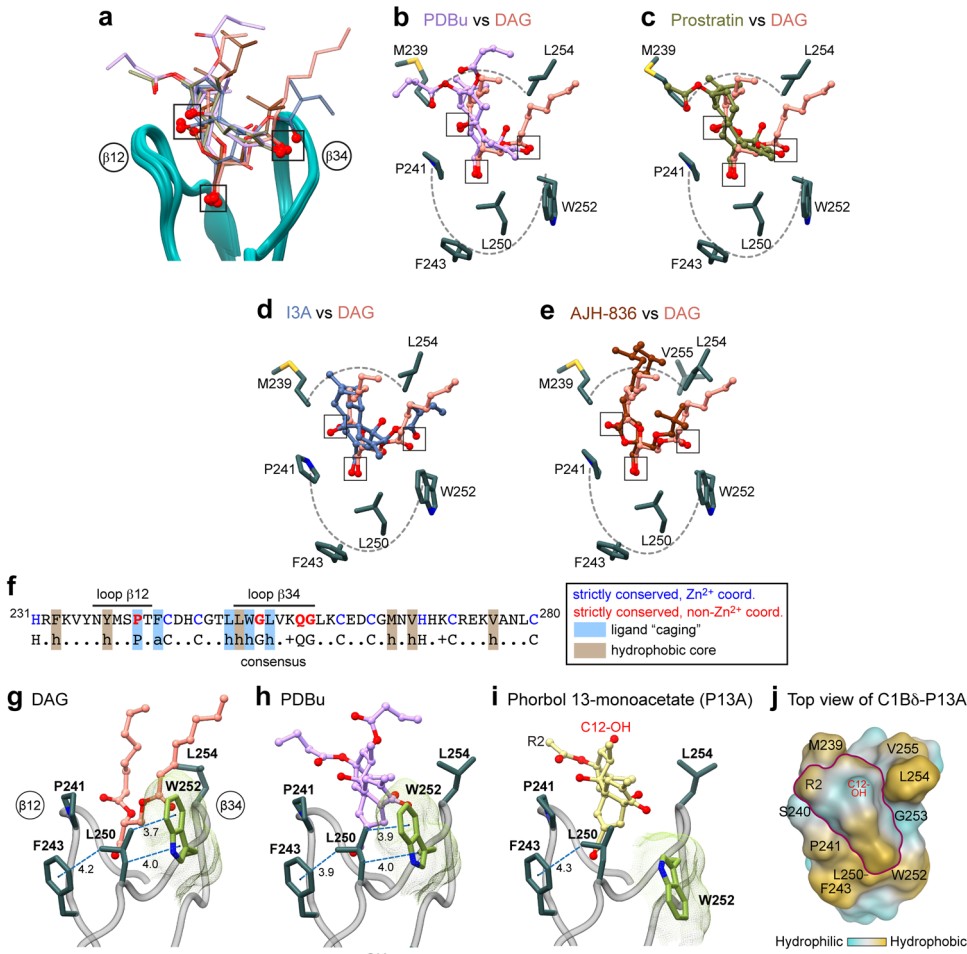

**Fig. 6 Structural analysis of C1Bδ-agonist complexes identifies three key oxygen-containing groups and the roles of conserved hydrophobic residues.** **a** Loop region of the backbone-superimposed C1Bδ complexes (pairwise r.m.s.d. <0.6 Å relative to chain 5 of the "sn-1" DAG complex). Oxygen-containing functional groups involved in the interactions with C1Bδ are highlighted by squares. **b–e** Pairwise comparison of the binding poses of **b** PDBu; **c** prostratin; **d** ingenol-3-angelate; and **e** AJH-836 relative to that of DAG in the binding groove. Hydrophobic sidechains that envelope the ligands and form the rim of the membrane-binding region are also shown. **f** Amino acid sequence of C1Bδ and the consensus sequence of DAG-sensitive C1 domains. **g–i** A subset of conserved hydrophobic residues that form a "cage"-like arrangement around the ligands, with a potential to form CH–π interactions in addition to the apolar contacts. The rotameric flip of Trp252, illustrated using DAG (**g**) and PDBu (**h**) complexes, is essential for creating a contiguous hydrophobic surface. **i** The Trp252 sidechain remains in its apo-state rotameric conformation in the C1Bδ-P13A complex that was crystallized in the absence of lipids/detergents. **j** Top view of the C1Bδ-P13A loop region showing the contribution of the C12-OH group to the hydrophilic character of the P13A "cap".

residues, provide unprecedented insight into the structural basis of DAG sensing. These also provide key information for guiding the design of therapeutic agonists that selectively target proteins within the DAG effector family. The critical advance that resolved the Gordian knot of C1-agonist crystallization was the inclusion of the membrane-mimicking agents into the system that provides the hydrophobic environment required to support high-affinity interactions. This general strategy now paves the way for the structural characterization of other C1-agonist complexes.

## Methods

**Expression, purification, and isotope enrichment of C1Bδ.** The cDNA segment encoding the C1B domain of protein kinase C (PKC) δ isoenzyme from *Rattus norvegicus* (amino acids 229–281) was sub-cloned into pET SUMO expression vector (Invitrogen). The His$_6$-SUMO-C1Bδ fusion protein was expressed in *Escherichia coli* BL21(DE3) Rosetta2 cells (Millipore Sigma). For the natural abundance preparations, the cells were grown in LB broth until OD$_{600}$ = 0.6, followed by the induction of protein expression with 0.5 mM IPTG at 18 °C for 16 h. For the isotopically enriched C1Bδ preparations, we used the resuspension method[41] in the M9 minimal medium supplemented with $^{15}$NH$_4$Cl and D-$^{13}$C$_6$-glucose as nitrogen and carbon sources, respectively. To obtain [~80% $^2$H, U-$^{15}$N,$^{13}$C]-enriched C1Bδ, M9 was prepared in 100% D$_2$O and additionally

supplemented with 1 g of $^{15}$N,$^{13}$C,$^2$H ISOGRO® (Sigma). Cell harvesting, lysis, and C1Bδ purification were carried out as previously described[21,22]. The purified protein was stored at 4 °C in the "storage buffer" comprising 50 mM MES at pH 6.5, 150 mM KCl, and 1 mM TCEP, until further use. For NMR experiments, C1Bδ was exchanged into an "NMR buffer" composed of 20 mM d$_4$-Imidazole at pH 6.5, 50 mM KCl, 0.1 mM TCEP, 0.02% NaN$_3$, and 8% D$_2$O.

**Preparation of isotropically tumbling bicelles.** Chloroform solutions of long-chain 1,2-dimyristoyl-*sn*-glycero-3-phosphocholine (DMPC) and short-chain 1,2-dihexanoyl-*sn*-glycero-3-phosphocholine (DHPC) (Avanti Polar Lipids), or their deuterated versions d$_{54}$-DMPC (Avanti Polar Lipids) and d$_{40}$-DHPC (Cambridge Isotope Laboratories), were aliquoted and dried extensively under vacuum. The bicelles of q = 0.5 (defined by the DMPC to DHPC molar ratio of 1:2) were prepared by suspending the dried lipid films in the NMR buffer, as previously described[42]. Additional lipid components: 1,2-dimyristoyl-*sn*-glycero-3-phospho-L-serine (DMPS) and di-octanoyl-*sn*-1,2-glycerol (DAG) were incorporated for all DAG-binding experiments, to produce the final molar ratios of DMPC:DMPS:DAG = 75:15:10. For the paramagnetic relaxation enhancement (PRE) measurements, 1-palmitoyl-2-stearoyl-(14-doxyl)-*sn*-glycero-3-phosphocholine (14-doxyl PC) was incorporated into bicelles to give on average ~1 molecule per leaflet. DMPS, DAG, and 14-doxyl PC were obtained from Avanti Polar Lipids. The lipid concentrations of final bicelle preparations were measured using the phosphate determination assay[43].

**NMR spectroscopy**. All NMR experiments were carried out at 25 °C (calibrated with $d_4$-methanol) on the Avance III HD NMR spectrometer (Bruker Biospin), operating at a $^1$H Larmor frequency of 800 MHz (18.8 T) and equipped with a cryogenically cooled probe. The data were processed with NMRPipe[44] and analyzed with NMRFAM-Sparky[45]. The backbone amide ($^{15}$NH) and methyl ($^{13}$CH$_3$) resonance assignments of C1Bδ were obtained from our previous work[22] and the BMRB entry 17112[46].

**NMR detection of C1Bδ-agonist complex formation in bicelles**. The ternary C1Bδ-agonist-bicelle complexes were assembled in the "NMR buffer" by combining solutions of the isotopically enriched protein, bicelles, and agonists. DAG was incorporated at the bicelle preparation stage (vide supra). The 30–40 mM ligand stock solutions were prepared in $d_6$-DMSO from the crystalline solids (Phorbol-12,13-dibutyrate, Prostratin, and Ingenol-3-angelate, all from Sigma-Aldrich®; and AJH-836, custom synthesized in Prof. Jeewoo Lee's laboratory). The samples for the [$^{15}$N,$^1$H] HSQC ([$^{13}$C,$^1$H] HSQC) experiments contained 0.4 mM [U-$^{15}$N,$^{13}$C]-enriched C1Bδ and 100 mM bicelles (0.3 mM [~80% $^2$H, U-$^{15}$N,$^{13}$C]-enriched C1Bδ and 80 mM deuterated bicelles). At these concentrations, the bicelle particles are approximately equimolar to protein. The protein-to-ligand molar ratios were 1:8 (DAG), 1:1.2 (PDBu/Prostratin/Ingenol-3-angelate), and 1:6 (AJH-836). The residue-specific chemical shift perturbations (CSPs, Δ) between the apo and agonist-bound C1Bδ were calculated using the following equation:

$$\Delta = \sqrt{\Delta\delta_H^2 + (\alpha\Delta\delta_X)^2} \qquad (1)$$

where $\Delta\delta_H$ and $\Delta\delta_X$ are the chemical shift changes of $^1$H and X ($^{15}$N or $^{13}$C), respectively; and α = 0.152 ($^{15}$N) or 0.18 ($^{13}$C).

**PRE and NOESY experiments**. The residue-specific PRE values of $^1$H$_N$ resonances, $\Gamma_2$, were determined using relaxation experiments with [$^{15}$N,$^1$H] TROSY-HSQC detection. All data were collected in the interleaved manner, with a two-point (0 and 10 ms) relaxation delay scheme[47]. The diamagnetic $^1$H$_N$ transverse relaxation rate constants, $R_{2,dia}$, were obtained on the NMR sample comprising 0.4 mM [~80% $^2$H, U-$^{15}$N,$^{13}$C]-enriched C1Bδ, 100 mM (total lipid) bicelles, and 3.2 mM DAG. The paramagnetic sample was prepared by a 1-hr room-temperature incubation of the diamagnetic sample with a dry film of 14-doxyl PC, and subsequently used to obtain the $^1$H$_N$ $R_{2,para}$ values. The error was estimated using the r.m.s.d. of the base plane noise. The $\Gamma_2$ values were calculated using the following equation:

$$\Gamma_2 = R_{2,para} - R_{2,dia} \qquad (2)$$

3D $^{15}$N-edited NOESY-TROSY experiment was carried out with a mixing time of 120 ms on a sample containing 0.4 mM [~80% $^2$H, U-$^{15}$N,$^{13}$C]-enriched C1Bδ and 100 mM (total lipid) deuterated bicelles that contained 3.2 mM DAG. Intermolecular C1Bδ-DAG $^1$H–$^1$H NOEs were identified based on the available assignments of the protein amide resonances and characteristic $^1$H chemical shifts of the sn-1,2 stereoisomer of DAG. The DAG chemical shifts were obtained using the $^{13}$C-$^1$H HSQC spectrum of the 0.4 mM [$^{13}$C,C1-C3]-racemic DAG (custom synthesized by Avanti Polar Lipids) in 10 mM $d_{38}$-dodecylphosphocholine (DPC, Sigma) micelles, and matched the literature values[48].

**Crystallization of apo C1Bδ and its complexes with agonists**. Apo C1Bδ and its complexes with agonists were crystallized at 4 °C using a hanging-drop vapor-diffusion method. The protein was concentrated at 2–2.3 mM (13–15 mg/ml) in the "storage buffer". 200 mM stock solutions of DPC (Sigma) and DHPC micelles were prepared using the procedure described previously[21]. To form the C1Bδ-DAG complex, the reagents were combined at a molar ratio of C1Bδ:DAG:DPC = 1:1.2:10. The C1Bδ-DAG crystals were obtained in 2 days from the precipitant comprising 0.2 M ammonium acetate, 0.1 M sodium phosphate at pH 6.8, and 15% isopropanol. These crystals were used as seeds for the hanging drop containing C1Bδ:DAG:DPC = 1:1.2:5 and allowed to grow for 2 more days.

To form the C1Bδ-ligand complexes, the appropriate reagents were combined at a molar ratio of C1Bδ:Ligand:DHPC = 1:1.2:10, where the ligand was PDBu, Prostratin, Ingenol-3-angelate, or AJH-836. To achieve full saturation of C1Bδ with AJH-836, we used an additional condition with AJH-836 C1Bδ:AJH-836:DHPC = 1:1.5:10. For all C1Bδ-ligand complexes, the precipitant was 0.2 M ammonium acetate, 0.1 M sodium phosphate at pH 6.8, and 30% isopropanol. The same precipitant composition was used to crystallize apo C1Bδ. Crystals of the apo C1Bδ and its ligand complexes appeared either overnight (PDBu, Prostratin, and Ingenol-3-angelate complexes) or after ~1–2 days (AJH-836 complex).

Prior to data collection, the crystals were flash-frozen in the following cryo-protectant solutions: (i) 15% sucrose and 25% PEG 4000 in 0.1 M sodium phosphate at pH 7.2 (apo C1Bδ and its complexes with PDBu, Prostratin, and Ingenol-3-angelate); and (ii) 20% MPD (Hampton Research) in mother liquor (C1Bδ complexes with DAG and AJH-836).

**Data collection, processing, and model building of C1Bδ-agonist complexes**. For the apo structure and all the ligands except AJH-836 the data were collected on a home source Cu k-alpha X-ray generator. The data were indexed, scaled, and

integrated by PROTEUM software[49]. For the C1Bδ-DAG complex crystal, the data were collected at Argonne National Lab APS synchrotron, beamline 23ID, and for the C1Bδ-AJH-836 complex crystal—at ALS synchrotron at Berkley, beamline BL502. The data were indexed, integrated, and scaled by the beamline auto-processing pipeline (XDS[50], POINTLESS[51], and AIMLESS[52] software packages). Structures were solved by molecular replacement, using the PDB entry 1PTQ as a search model[17]. This was followed by iterative cycles of refinement with PHENIX.REFINE and manual building in COOT[53,54]. Polder omit maps were generated using PHENIX[55]. Ligands were created using ELBOW.BUILDER and JLIGAND[56,57]. Structural analyses were carried out using UCSF Chimera[58], CCG Molecular Operating Environment (MOE)[59], and LigPlot+[60].

Although the diffraction data of the C1Bδ-DAG complex could be indexed and scaled in the F23 space group, the structure was solved in the lower symmetry H3 space group because of the non-uniform lipid molecules in the solvent channels. We have built only the lipids for which well-ordered electron density of head groups was present. The peripheral lipids and detergents are less ordered, with average B factors of 71 and 52 Å$^2$ for DAG and DPC, respectively (the average protein B factor is 36 Å$^2$). However, there are likely more lipids and/or detergents in the solvent channels, as evidenced by the multitude of positive difference electron density peaks that are larger than water, some reaching across the symmetry axis.

The coordinates of all structures were deposited in the Protein Data Bank. The accession numbers and statistics are given in Supplementary Tables 1 and 2.

**Reporting summary**. Further information on research design is available in the Nature Research Reporting Summary linked to this article.

## Data availability

The data that support this study are available from the corresponding author upon reasonable request. The atomic coordinates and structure factors are deposited in the PDB (https://www.rcsb.org) under the accession codes 7KND (apo), 7L92 (DAG), 7LEO (AJH-836), 7LF3 (AJH-836), 7KNJ (PDBu), 7LCB (Prostratin), and 7KO6 (I3A).

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

## Acknowledgements

This research was supported by the U.S. National Institute of Health grant R01 GM108998 and Texas A&M institutional funds to T.I.I., and Welch Foundation grant A-0015 to J.C.S. We acknowledge the beamline scientists of 19 and 23 ID stations of the Advanced Photon Source synchrotron at Argonne National Laboratory and Berkeley Center for Structural Biology scientists of beamline 502 at the Advanced Light Source at Lawrence Berkeley National Laboratory. We are grateful to Dr. Vytas A. Bankaitis for critical reading of the manuscript and suggestions.

## Author contributions

S.S.K. and T.I.I. conceptualized the work, developed the crystallization approach, and designed NMR experiments. S.S.K. prepared all samples, crystallized the complexes, and collected and analyzed NMR data. I.V.K. and S.S.K. collected the X-ray diffraction data. I.V.K. built and refined the models. S.S.K., I.V.K., J.C.S., and T.I.I. analyzed the structural data. J.A. and J.L. synthesized and provided the AJH-836 compound. T.I.I. and S.S.K. wrote the manuscript with input from all coauthors. T.I.I. supervised the project.

## Competing interests

S.S.K. and T.I.I. declare the existence of a potential financial interest due to a provisional patent application (Application No. 63/239,986) pending with the United States Patent and Trademark Office, with the specific aspect of this manuscript covered in the application being the method of crystallization of C1 domains complexed to ligands (T.I.I. and S.S.K. are listed as inventors and applicants, under obligation to assign to Texas A&M University). The remaining authors declare no competing interests.

**Additional information**

