## [Peer Review File · Nature Communications]

Structural anatomy of Protein Kinase C C1 domain interactions with diacylglycerol and other agonistsReviewers' Comments:

Reviewer #1:

Remarks to the Author:

This manuscript presents an exciting and novel study of how PKC-C1 domain interacts with DAG and other agonists. The authors have reported the first high-resolution X-ray structure of a C1B δ domain in complex with DAG and with four different PKC agonists. These agonists produce different biological readouts and are particularly important from therapeutic perspective. The C1B δ domains form an intricate network within the lattice: the unit cell contains 72 DAG-complexed C1B δ chains and 18/54 DAG/DPC molecules that peripherally associate with the protein surface; the asymmetric unit comprises 8 C1B δ protein chains with 8 DAG molecules captured within a well-defined groove, and 2/6 peripheral DAG/DPC molecules. To complete the story and to eliminate the possibility that the interactions highlighted were crystallization artifacts not found in nature, authors used solution NMR to study the interactions in question using isotropic bicelles as membrane mimetics.

As an NMR expert myself I can confirm that NMR part (chemical shifts mapping, PREs, inter-molecular NOEs) is solid: experiments are logically designed, clearly defined, and performed with necessary attention to details. In my opinion, these data conclusively proves that high-resolution structures of five C1B δ -ligand complexes are natural, particularly with regard to the arrangement of the ligand hydrophobic substituents within the binding groove and assignment of specific functional roles to the key C1B δ residues. Altogether the manuscript provides unprecedented insight into the structural basis of DAG sensing and is worth publishing in Nature Communications.

Reviewer #2:

Remarks to the Author:

This work promises to be an important contribution to the field, and represents a tour-de-force of structural biology. The work combines an array of methodologies, some novel, including membrane protein solubilization, crystallization, X-ray crystallography and NMR. The findings provide new insights into the C1 diacylglycerol binding domain central to the regulation of key signaling proteins and pathways, including PKC isoforms and their signaling circuits. These insights include high resolution crystal structures of the apo PKCdelta C1B domain and its complexes with native activator (DAG) and four other activators. The work will be suitable for publication in Nature Communications following adequate revisions to address several major points.

Major points:

1. As the authors note, a previous publication from Hurley et al first described the crystal structure of apo PKCdelta C1B domain and its complex with the weak activator phorbol-13-acetate (ref 19), and the authors used this information to solve the new crystal structures presented herein by molecular replacement. However, the MS fails to adequately consider the previous co-complex structure in its model for C1 regulation. In each of the 5 new activator-C1B co-complex structures, W252 has flipped from its "inactive" conformation observed in the previous co-complex, in which the indole faces away from the activator binding loops, to adopt its "active" conformation facing towards the activator binding loops. The most relevant comparison is that of the prior C1B co-complex structure (with phorbol-13-acetate) and the new C1B complex with 12-deoxyphorbol-13-acetate (Prostratin), since these two ligands differ only with respect to a single hydroxyl at the 12 position. The failure of W252 to flip into the active conformation in the previous structure could be due to destabilization of the active conformation of the indole by the 12-hydroxyl. Alternatively, the membrane-mimicking micelle of the new Prostratin complex could stabilize the active W252 conformation. Or both. The authors should discuss these points in detail and develop a wholistic model that includes the prior co-complex structure.

2. A related point - is the new apo structure identical to the prior apo structure 1PTQ? If not, does this reflect different detergent or micelle environments, etc? Please discuss.

3. This strong manuscript would be further strengthened by inclusion of a structure of C1B with PMA (phorbol-12-myristate-13-acetate). PMA is a stronger PKC activator than any of those employed (see ref 37) and is thus preferred in many research applications. Inclusion of a C1B-PMA co-complex may be beyond the current scope. Either way, the greater strength of PMA as an activator has important implications for the activation model that should be discussed. The efficacy of PMA as an activator argues that a longer acyl chain at the 12 position stabilizes the active state by a) stabilizing the active conformation of W252 and the beta 34 loop, and/or b) increasing the affinity of the C1B-activator complex for the bilayer due to hydrophobic interactions with the longer acyl chain. Currently the manuscript focuses on the interactions between acyl chains at the 12 and 13 phorbol positions and the protein, and their impacts on the hydrophobic surface of the protein. However, in addition to generating a continuous hydrophobic surface, a substantial acyl chain can expand the hydrophobic surface area and generate significant, direct acyl chain-membrane interactions that stabilize the membrane-bound complex. The MS fails to point out the importance of such direct activator-membrane interactions. PKC activation requires both binding of C1B to the activator, and C1B binding to the bilayer to recruit this auto-inhibitory domain away from the catalytic domain, representing a coupled binding equilibrium stabilized by both activator-protein and activator-membrane interactions. See PMID 24559055 and 25075641.

4. The impressive NMR lipid-to-protein PRE data (Fig 3b) are obtained in a bicelle environment that better mimics the native lipid bilayer than the micelle environment employed for crystallization. Is the observed PRE pattern fully consistent with the C1B conformation in the crystal structure, or are there inconsistencies that could reveal interesting differences between the C1B conformation in the bicelle and micelle environments?

Minor points:

5. Previous single molecule diffusion studies and atomistic molecular dynamics simulations have both indicated that C1B of PKC α sits in a shallower, lower friction membrane position when bound to a phorbol activator (PMA), compared to a DAG activator. See PMID 25075641. Perhaps the authors could provide NMR lipid-to-protein PRE data to shed further light on the different membrane docking geometries of C1B bound to phorbol and DAG activators? Of course, inclusion of such data may be beyond the current scope.

6. Extended Fig 8 is very useful and would be further strengthened by addition of the apo and C1B-DAG structures for full comparison.

RESPONSE TO REVIEWERS

We thank the reviewers for their comments and the opportunity to improve the clarity and depth of the manuscript. Reviewer 1 did not raise any concerns. The response to Reviewer 2 is given below.

Reviewer #2

Major points:

1. As the authors note, a previous publication from Hurley et al first described the crystal structure of apo PKCdelta C1B domain and its complex with the weak activator phorbol-13-acetate (ref 19), and the authors used this information to solve the new crystal structures presented herein by molecular replacement. However, the MS fails to adequately consider the previous co-complex structure in its model for C1 regulation. In each of the 5 new activator-C1B co-complex structures, W252 has flipped from its "inactive" conformation observed in the previous co-complex, in which the indole faces away from the activator binding loops, to adopt its "active" conformation facing towards the activator binding loops. The most relevant comparison is that of the prior C1B co-complex structure (with phorbol-13-acetate) and the new C1B complex with 12-deoxyphorbol-13-acetate (Prostratin), since these two ligands differ only with respect to a single hydroxyl at the 12 position. The failure of W252 to flip into the active conformation in the previous structure could be due to destabilization of the active conformation of the indole by the 12-hydroxyl. Alternatively, the membrane-mimicking micelle of the new Prostratin complex could stabilize the active W252 conformation. Or both. The authors should discuss these points in detail and develop a wholistic model that includes the prior co-complex structure.

Response: Done

We have included the discussion of the previous structure of the C1B δ -P13A complex in our model for C1 regulation. We have added:

- (1) **panel j to Extended Figure 9** that shows the properties of the C1B δ -P13A complex
- (2) Description of the holistic model that includes the prior C1B δ -P13A complex (highlighted on page 16 of the main manuscript; **PMID 24559055 is cited as ref. 41**):

“Given that all C1 complexes reported in this work are with potent PKC agonists, we posit that the reorientation of the Trp252 in C1B δ (**Fig. 2a**) is an essential aspect of the overall mechanism of membrane recruitment and agonist capture. Indeed, our previous work suggests that the Trp252 sidechain reorients towards the membrane binding loops upon initial partitioning of C1B δ into the hydrophobic environment prior to the agonist binding²⁴. Once this “pre-DAG” C1 complex⁴¹ is formed, the Trp sidechain plays an important role in the formation of the ligand-binding site through the completion of the hydrophobic “cage” (**Ext. Fig 9, g-h**). Therefore, the structural change associated with the Trp252 flip underlies two processes that take place in the hydrophobic environment – membrane partitioning and binding of the membrane-embedded ligand. Consistent with this notion, no rotameric flip was observed in the previously determined structure of the C1B δ -phorbol-13-acetate (P13A) complex that was obtained under crystallization conditions lacking membrane mimics (**Ext. Fig 9i**)¹⁹. P13A is an extremely weak agonist of PKC^{42,43} that differs from Prostratin with respect to a single hydroxyl group at the C-12 position. This OH group is ~10 Å away from Trp252 and is unlikely to influence the sidechain conformation directly (**Ext. Fig 9i**). Rather, it imparts significant hydrophilic character onto the “cap” formed by the ligand over the loop regions of C1B δ (**Ext. Fig 9j**). Given the relative hydrophilicity of the ligand (logP of P13A is 0.2, as compared with Prostratin’s 0.8), and of the complex itself, there is no thermodynamic incentive for the former to partition into the membranes in a process that involves the Trp252 sidechain reorientation.”

2. A related point - is the new apo structure identical to the prior apo structure IPTQ? If not, does this reflect different detergent or micelle environments, etc? Please discuss.

Response: Done

We have added the following text to the main manuscript (highlighted on page 5 of the main manuscript):

“We crystallized apo C1B δ under conditions similar to those used for the C1B δ -DAG complex (but without detergent) for a direct comparison. We found the structure to be identical to the previously reported apo structure (IPTQ¹⁹) with a backbone r.m.s.d. of 0.4 Å.”

3. *This strong manuscript would be further strengthened by inclusion of a structure of C1B with PMA (phorbol-12-myristate-13-acetate). PMA is a stronger PKC activator than any of those employed (see ref 37) and is thus preferred in many research applications. Inclusion of a C1B-PMA co-complex may be beyond the current scope. Either way, the greater strength of PMA as an activator has important implications for the activation model that should be discussed. The efficacy of PMA as an activator argues that a longer acyl chain at the 12 position stabilizes the active state by a) stabilizing the active conformation of W252 and the beta 34 loop, and/or b) increasing the affinity of the C1B-activator complex for the bilayer due to hydrophobic interactions with the longer acyl chain. Currently the manuscript focuses on the interactions between acyl chains at the 12 and 13 phorbol positions and the protein, and their impacts on the hydrophobic surface of the protein. However, in addition to generating a continuous hydrophobic surface, a substantial acyl chain can expand the hydrophobic surface area and generate significant, direct acyl chain-membrane interactions that stabilize the membrane-bound complex. The MS fails to point out the importance of such direct activator-membrane interactions. PKC activation requires both binding of C1B to the activator, and C1B binding to the bilayer to recruit this auto-inhibitory domain away from the catalytic domain, representing a coupled binding equilibrium stabilized by both activator-protein and activator-membrane interactions. See PMID 24559055 and 25075641.*

Response: Done

We have included the discussion of the significance of direct activator-membrane interactions (highlighted on page 14 of the main manuscript; **PMID 25075641 is cited as ref. 40**):

“The increased potencies³⁷ of 12,13-di- and 12-mono-esters with hydrophobic substituents (e.g., PDBu and phorbol 12-myristate 13-acetate (PMA)) relative to 12-deoxyphorbol esters (e.g., Prostratin) can be ascribed to two factors. First, the C-12 R1 group complements the hydrophobic rim of loop β 34 that is the primary driver of C1 membrane binding. Second, both R1 and R2 groups are involved in direct interactions with lipids, and thereby contribute to the stabilization of the membrane complex. Molecular dynamics simulations of the C1-ligand complexes suggest that both the chemical identity of the R1 group³⁹ and of the ligand itself (PMA vs. DAG)⁴⁰ influence the depth of C1 membrane insertion.”

The second point raised by the reviewer is the structural characterization of the PMA-complexed C1 domain. This constitutes a separate and major project in itself due to the unique challenges PMA presents as a C1 ligand. At least half of its 14-carbon myristoyl tail is expected to protrude out of the C1 protein pocket. Our direct experience on these matters teaches us that finding a membrane-mimicking environment that would accommodate such an arrangement, and yet be compatible with crystal formation, requires an extensive search and optimization of the crystallization conditions. The past several decades of unsuccessful attempts at crystallizing C1-DAG complexes further testify to the difficulties associated with satisfying this suggestion in any reasonable time frame. The time and

uncertainties involved in attempting to obtain and then include such a crystal ask for work that is beyond the scope of this manuscript.

4. The impressive NMR lipid-to-protein PRE data (Fig 3b) are obtained in a bicelle environment that better mimics the native lipid bilayer than the micelle environment employed for crystallization. Is the observed PRE pattern fully consistent with the C1B conformation in the crystal structure, or are there inconsistencies that could reveal interesting differences between the C1B conformation in the bicelle and micelle environments?

Response:

The nature of the lipid-to-protein PREs is such that they do not report directly on the conformation of the protein. They report only on the distances between protein amide groups and the paramagnetic lipid (i.e. insertion depth). Direct evidence that the crystallized C1B-DAG complex is similar to the C1B-DAG complex in bicelles is provided by the **NOE data of Figure 2f**. The NOE patterns predicted from the crystalline “sn-1” C1B-DAG complex match the experimental NOE data obtained for the C1B-DAG complex in bicelles. The C1B-DAG PRE patterns observed in bicelles (this work) and micelles (our previous work, ref 24) are similar, with loop β 34 showing more extensive broadening than loop β 12.

Minor points:

5. Previous single molecule diffusion studies and atomistic molecular dynamics simulations have both indicated that C1B of PKC α sits in a shallower, lower friction membrane position when bound to a phorbol activator (PMA), compared to a DAG activator. See PMID 25075641. Perhaps the authors could provide NMR lipid-to-protein PRE data to shed further light on the different membrane docking geometries of C1B bound to phorbol and DAG activators? Of course, inclusion of such data may be beyond the current scope.

Response:

The reviewer raises an interesting point. We intend to pursue the PRE studies of the C1B-ligand complexes in the future to understand how different PKC agonists influence the C1-membrane interactions. This merits a large study of its own, and we believe that work falls outside the scope of this manuscript.

6. Extended Fig 8 is very useful and would be further strengthened by addition of the apo and C1B-DAG structures for full comparison.

Response: Done.

We have added two extra panels, **d** (apo structure) and **e,f** (“sn-1” and “sn-2” DAG structures), to the **Extended Figure 8**.